# *Vernonia calvoana* Shows Promise towards the Treatment of Ovarian Cancer

**DOI:** 10.3390/ijms21124429

**Published:** 2020-06-22

**Authors:** Ariane T. Mbemi, Jennifer N. Sims, Clement G. Yedjou, Felicite K. Noubissi, Christian R. Gomez, Paul B. Tchounwou

**Affiliations:** 1Natural Chemotherapeutics Research Laboratory, NIH/NIMHD RCMI-Center for Environmental Health, College of Science, Engineering and Technology, Jackson State University, 1400 Lynch Street, Jackson, MS 39217, USA; ariane.t.mbemi@jsums.edu; 2School of Public Health, Jackson State University, Jackson Medical Mall-Thad Cochran Center, 350 West Woodrow Wilson Avenue, Jackson, MS 39213, USA; jennifer.n.sims@jsums.edu; 3Department of Biological Sciences, College of Science and Technology, Florida Agricultural and Mechanical University, 1610 S. Martin Luther King Blvd, Tallahassee, FL 32307, USA; 4Department of Biology, College of Science, Engineering and Technology, Jackson State University, 1400 Lynch Street, Jackson, MS 39217, USA; felicite.noubissi_kamdem@jsums.edu; 5Departments of Pathology and Radiation Oncology, Center for Clinical and Translational Science, University of Mississippi Medical Center, 2500 N. State St., Jackson, MS 39216, USA; CRGomez@umc.edu

**Keywords:** *Vernonia calvoana*, OVCAR-3 cells, cell viability, oxidative stress, DNA damage, cell cycle

## Abstract

The treatment for ovarian cancers includes chemotherapies which use drugs such as cisplatin, paclitaxel, carboplatin, platinum, taxanes, or their combination, and other molecular target therapies. However, these current therapies are often accompanied with side effects. *Vernonia calvoana* (VC) is a valuable edible medicinal plant that is widespread in West Africa. In vitro data in our lab demonstrated that VC crude extract inhibits human ovarian cancer cells in a dose-dependent manner, suggesting its antitumor activity. From the VC crude extract, we have generated 10 fractions and VC fraction 7 (F7) appears to show the highest antitumor activity towards ovarian cancer cells. However, the mechanisms by which VC F7 exerts its antitumor activity in cancer cells remain largely unknown. We hypothesized that VC F7 inhibits cell proliferation and induces DNA damage and cell cycle arrest in ovarian cells through oxidative stress. To test our hypothesis, we extracted and fractionated VC leaves. The effects of VC F7 were tested in OVCAR-3 cells. Viability was assessed by the means of MTS assay. Cell morphology was analyzed by acridine orange and propidium iodide (AO/PI) dye using a fluorescent microscope. Oxidative stress biomarkers were evaluated by the means of lipid peroxidation, catalase, and glutathione peroxidase assays, respectively. The degree of DNA damage was assessed by comet assay. Cell cycle distribution was assessed by flow cytometry. Data generated from the MTS assay demonstrated that VC F7 inhibits the growth of OVCAR-3 cells in a dose-dependent manner, showing a gradual increase in the loss of viability in VC F7-treated cells. Data obtained from the AO/PI dye assessment revealed morphological alterations and exhibited characteristics such as loss of cellular membrane integrity, cell shrinkage, cell membrane damage, organelle breakdown, and detachment from the culture plate. We observed a significant increase (*p* < 0.05) in the levels of malondialdhyde (MDA) production in treated cells compared to the control. A gradual decrease in both catalase and glutathione peroxidase activities were observed in the treated cells compared to the control. Data obtained from the comet assay showed a significant increase (*p* < 0.05) in the percentages of DNA cleavage and comet tail length. The results of the flow cytometry analysis indicated VC F7 treatment caused cell cycle arrest at the S-phase checkpoint. Taken together, our results demonstrate that VC F7 exerts its anticancer activity by inhibiting cell proliferation, inducing DNA damage, and causing cell cycle arrest through oxidative stress in OVAR-3 cells. This finding suggests that VC F7 may be a potential alternative dietary agent for the prevention and/or treatment of ovarian cancer.

## 1. Introduction

Ovarian cancer is classified as the leading cause of death in gynecological cancer among women [1,2]. Family history of breast or ovarian cancer is the strongest risk factor for developing ovarian cancer [3]. The American College of Obstetricians and Gynecologists and the Society of Gynecologic Oncology recommend that females with BRCA1 and/or BRCA2 mutations should consider the use of oral contraceptives which may reduce the risk of developing ovarian cancer by approximately 50% among high-risk females [4,5,6,7]. Due to the location of the ovaries in the female reproductive system, ovarian cancer is considered a “silent killer”, and over 70% of cases are diagnosed at the advanced stages [8,9]. Last year, there was an approximate of 22,240 new cases of ovarian cancer diagnosed and about 14,070 ovarian cancer deaths in North America [10]. Ovarian cancer accounts for about 2.5% of all malignancies among women worldwide. However, 5% of female cancer deaths are attributed to low survival rates, largely due to late stage diagnoses [11].

The current first line of treatment for ovarian cancers are chemotherapies which use drugs such as cisplatin, paclitaxel, carboplatin, platinum, taxanes, or their combination, and other molecular target therapies [10,12]. However, these therapies for ovarian cancer are usually accompanied with side effects such as hair loss, loss of appetite, and infertility [13]. About 70% of the patients diagnosed with recurring ovarian cancer will die within five years of their diagnosis [14]. Medicinal plants are gaining special consideration and value in the discovery for ovarian cancer drugs. *Vernonia calvoana* (VC), commonly called sweet bitter leaf in English, is an Asteraceae that has been widely consumed as vegetable and used to treat diseases such as diabetes, measles, tuberculosis, hyperlipidemia, and women infertility in many Africa countries [15,16]. Scientific reports indicated that the leaves of VC contain phytochemicals such as flavonoids, which are good antioxidants [17,18,19]. In addition, scientific reports indicated that VC possesses the hepatoprotective effect and hypolipidemic and antidiabetic activities [20]. Testing the medicinal property of *Vernonia amygdalina*, a member of the same genus as *Vernonia calvoana,* we demonstrated in our lab that this medicinal plant possesses anticancer activity potential against human breast cancer cells [21,22]. Other *Vernonia* species including *Vernonia divaricate* and *Vernonia amygdalina* act as potential anticancer agents that inhibit the proliferation of HL-60 cells, MCF-7 cells and PC-3 cells [23,24]. Although VC have been used traditionally to treat many illnesses, the mechanisms by which it exerts its antitumor activity in OVCAR-3 cells remain largely unknown. Therefore, our objective was to test the therapeutic efficacy of the most bioactive compound of VC against ovarian cancer.

## 2. Results

### 2.1. Antiproliferative Effect

The results showed that VC F7 treatment significantly decreased the viability of OVCAR-3 cells (Figure 1). As seen in Figure 1, the percentages of viability of OVCAR-3 cells treated with VC F7 upon 48 h were 100 ± 5.73%, 63 ± 2.78%, 43 ± 1.27 %, 31 ± 0.92% in 0, 8, 16, 32 µg/mL, respectively. These results revealed that the OVCAR-3 cells are more sensitive to VC F7 treatment with an estimated inhibition dose (IC_50_) equal to 18.56 µg/mL. These data showed that VC F7 caused growth arrest of OVCAR-3 cells, suggesting its potential as an anticancer agent.

### 2.2. Morphological Changes

To confirm the antiproliferative effect of VC F7 on OVCAR-3 cells, we examined the cell morphology by acridine orange/propidium iodide (AO/PI) double staining assay. We observed that VC F7 inhibits the proliferation of OVCAR-3 cells in a dose-dependent manner (Figure 2). As seen in Figure 2, there is a strong dose–response relationship in regard to VC F7 treatment, showing a significant increase in the percentage of dead cells compared to the percentage of live cells in the control. The control (0 µg/mL) OVCAR-3 cells display a normal round shape and remain firmly attached to the culture plate. Meanwhile, cells treated with VC F7 resulted in morphological alterations and exhibited characteristics such as loss of cellular membrane integrity, cell shrinkage, cell membrane damage, organelle breakdown, and detachment from the culture plate. Acridine orange/propidium iodide is a rapid, sensitive, and successful method to examine cellular morphology, live and dead cells, apoptosis and/or necrosis.

### 2.3. Induction of Oxidative Stress

To test whether oxidative stress play a role in VC F7 inducing the antiproliferative effect against ovarian cancer cells, we measured the levels of lipid peroxidation, catalase, and glutathione peroxidase in OVCAR-3 cells. Our result obtained from lipid peroxidation assay showed a significant (*p* < 0.05) increase in the production of malondialdehyde (a by-product of lipid peroxidation and biomarker of oxidative stress) in VC F7-treated cells compared to the control (Figure 3). Upon 48 h of treatment, the MDA values were 0.044 ± 0.058, 0.140 ± 0.034, 0.209 ± 0.010, and 0.324 ± 0.017 nmol in 0, 8, 16, and 32 µg/mL of VC F7, respectively.

To further understand the ability of VC F7 to induce oxidative stress in OVCAR-3 cells, we determined the activity of catalase. Data generated from catalase assay demonstrated that VC F7 slightly decreased the activity of catalase at 8 and 16 µg/mL of treatment. When cells were treated with VC F7 at 32 μg/mL, catalase activity showed a significant decrease (*p* < 0.05) compared to the control. The catalase activities were found to be 0.14 ± 0.07853, 0.13 ± 0.001, 0.13 ± 0.0001, and 0.048 ± 0.000416 n/mol in 0, 8, 16, and 32 µg/mL of VC F7, respectively (Figure 4).

To confirm our observations with the catalase activities, we performed a glutathione peroxidase assay. Data generated from the glutathione peroxidase assay showed a gradual decrease in glutathione peroxidase activity in OVCAR-3 cells treated with VC F7 compared to the vehicle control (Figure 5).

### 2.4. Induction of DNA Damage

To assess the ability of VC F7 to induce DNA damage in OVCAR-3 cells, we performed a comet assay. Data generated from this assay showed a gradual increase in the mean values of comet tail length, tail moment, and percentages of DNA cleavage of OVCAR-3 cells, with increasing doses of VC F7 (Figure 6). After 48 h of treatment, the percentages of DNA cleavage were computed to be 3.46 ± 0.043 %, 21 ± 0.953 %, 33.77 ± 0.529 %, and 51.46 ± 0.353% in the respective amounts of 0, 8, 16, and 32 µg/mL of VC F7 (Figure 7A). In the same order, the mean comet tail lengths computed were 7.47 ± 0.16, 27 ± 0.23, 56.44 ± 0.53, and 69 ± 0.45 µM in the respective amounts of 0, 8, 16, and 32 µg/mL of VC F7 (Figure 7B).

### 2.5. Induction of Cell Cycle Arrest

Here, our aim was to analyze the effects of VC F7 on cell cycle phases and find out at which phase/stage ceases the cycle. Treatment of OVCAR3 cells with VC F7 significantly increases cell population within the S-phase and significantly reduces it in the G2/M-phase (Figure 8 and Figure 9), indicating that VC F7 treatment causes cell cycle arrest at the S-phase checkpoint. The cell cycle arrest at the S-phase implies that the cell is unable to duplicate its DNA. Figure 8 shows the percentage of cells at the G0/G1, S, and G2/M regions. As seen in Figure 8, a cell cycle arrest at the S-phase is associated with a reduction in the G2/M-phase. Figure 9 shows representative dots plots and a histogram of cell cycle distribution of OVCAR-3 cells treated and untreated with VC-F7.

## 3. Discussion

### 3.1. Antiproliferative Effect

In the present study, we first evaluated the antiproliferative effect of VC F7 on OVCAR-3 cells by the means of MTS assay. Our results demonstrated that VC F7 significantly (*p* < 0.05) reduces the percentage of live cells in a dose-dependent manner, suggesting its antiproliferative effect against ovarian cancer (Figure 1). We further evaluated the antiproliferative effect of VC F7 and morphological changes of OVCAR-3 cells by the means of AO/PI assay. We observed that VC F7 gradually inhibits the proliferation of OVCAR-3 cells and causes morphological changes of these cells (Figure 2). As seen in Figure 2, the untreated OVCAR-3 cells display a normal round shape, have about 90 % confluency, and remain firmly attached to the culture plate. However, cells treated with VC F7 resulted in morphological alterations and exhibited characteristics such as loss of cellular membrane integrity, cell shrinkage, cell membrane damage, organelle breakdown, apoptotic bodies, and detachment from the culture plate [25]. We previously observed similar results while testing the antitumor activity of *Vernonia amygdalina*, a species of the same family as *Vernonia calvoana* [26]. In Cameroon, the leaves of *Vernonia amygdalina*, *Vernonia calvoana*, and *Vernonia amygdalina* Delile plants are used extensively as leaf vegetables and form a major constituent of a stew called ndole. The medicinal properties of these plants are well-documented and are commonly recommended by herbalists to patients in African countries for the treatment of headaches, stomach-aches [27], gastrointestinal tract problems [28], loss of appetite, breast milk enhancement in nursing mothers [29], bacterial infections, liver diseases, kidney problems [30], hypertension and diabetes [20,31], and cancer [24,32]. Animals use these medicinal plants to cure themselves and there is scientific evidence in which chimpanzees inhabiting the Mahale Mountains National Park in Tanzania have been observed chewing the pith of the leaves of *Vernonia Amygdalina*. Possible benefits include *Vernonia’s* ability to ward off parasites and to treat gastrointestinal tract infections [33]. Interestingly, the chimpanzees’ health condition gradually improved within a day and they resumed their normal activity. According to statistics, herbal and plant-derived medicines are the most frequently used therapies worldwide. A large number of people in developing countries depend on these natural remedies to maintain healthcare and a 38% increase in usage in the United States within the last decade of the 20th century alone has been reported [34,35]. It has been shown that natural medicinal plants work with the body to boost the immune system by killing unhealthy cells [36,37]. A previous scientific report indicated that VC possesses a hepatoprotective effect and hypolipidemic and antidiabetic activities [20].

### 3.2. Induction of Oxidative Stress

Given the effectiveness of VC F7 to inhibit the growth of OVCAR-3 cells, we hypothesized that the antiproferative effect of VC F7 may be mediated through oxidative stress. To test our hypothesis, we measured the levels of lipid peroxidation, catalase, and glutathione peroxidase in OVCAR-3 cells treated with VC F7. Oxidative stress plays an important role in cancer initiation and progression [38]. Our result obtained from lipid peroxidation assay showed a significant (*p <* 0.05) increase in the production of malondialdehyde (MDA—a by-product of lipid peroxidation and biomarker of oxidative stress) in VC F7-treated cells compared to the control. This finding is consistent with previous studies in our lab showing that *Vernonia amygdalina* Delile crude extract acts as a pro-oxidant in prostate cancer (PC-3) cells at a high concentration [39]. We also previously demonstrated that garlic extract significantly induced MDA production in the HL-60 human leukemia cell in a dose-dependent manner [39].

Our hypothesis that the antiproferative effect of VC F7 may be mediated through oxidative stress was also tested by catalase assay. Catalase is an enzyme that neutralizes the burden of H_2_O_2_ in cells by decomposing this molecule into water and oxygen. Catalase activity intercepts the oxidative damage that is triggered by high levels of H_2_O_2_. Studies have shown that the presence of high levels of H_2_O_2_ increases the speed of DNA mutation [40,41]. Our result revealed a gradual decrease in catalase activity in VC F7-treated cells when compared to the control.

To further understand the ability of VC F7 to induce oxidative stress in OVCAR-3 cells, we examined the activity of glutathione peroxidase. Glutathione peroxidase is localized in the cytosol and mitochondria, and research suggests that it may degrade low levels of hydrogen peroxide—one of the main ROS involved in arsenic-induced DNA damage [42,43]. Data demonstrated that VC F7 significantly (*p* < 0.05) decreases the levels of glutathione content. Depletion of glutathione level is associated with the early stage of the initiation of cell death [44,45].

Taken together, our results of oxidative stress are in agreement with a previous study showing that extracts of *Alhagi maurorum* increased the production of MDA levels, decreased the content of GSH, and decreased the activities of antioxidant enzymes including SOD, GPx, and GST in livers of STZ-induced diabetic rats [46].

### 3.3. Induction of DNA Damage

The ability of VC F7 to induce DNA damage in OVCAR-3 cells was determined by the means of comet assay. Our results showed that VC F7 induces DNA damage in OVCAR-3 cells in a dose-dependent manner. Untreated cells show low or no DNA migration, indicating that the DNA is intact and undamaged. However, cells treated with VC F7 revealed a gradual increase in DNA cleavage as well as an increase in comet tail length when compared to that of the untreated cells (control). To the best of our knowledge, no data are found in the literature regarding the genotoxic effect of VC in vitro or in vivo. We revealed for the first time that VC F7 induces DNA damage in OVCAR-3 cells, supporting its ability as a potential DNA-damaging anticancer agent effective against ovarian cancer. Working with other *Vernonia* species, previous reports from our laboratory demonstrated that in vitro *Vernonia amygdalina* treatment reduces cellular viability, that is, it induces DNA damage leading to apoptosis accompanied by secondary necrotic cells in human breast cancer (MCF-7) cells [21,22]. In another study, we showed that *Vernonia amygdalina* Delile induced DNA damage in human leukemia (HL-60) cells and human prostate cancer (PC-3) cells [24].

### 3.4. Induction of Cell Cycle Arrest

To assess the effects of VC F7 on cell cycle and population distribution, OVCAR-3 cells were stained with propidium iodide and analyzed by flow cytometry. We found that 48-h VC F7 treatment induced significant cell cycle arrest in the S-phase (*p* < 0.05) in comparison to untreated cells. The cell cycle arrest in the S-phase is a direct result of VC F7 inhibition of cell growth and induction of DNA damage in OVCAR-3 cells via oxidative stress (Figure 10). Consistent with our data, many natural products exhibit inhibitory effects on cancer cells via disruption of cell cycle progression. For example, studies showed that *Ganoderma* extract caused cell cycle arrest in cancer cells [47]. The growth inhibitory effect of celery seed extracts on human gastric cancer BGC-823 cells caused cell cycle arrest at the S-phase and decreased levels of cyclin A and CDK-2 [48,49]. The crude water extract of *Centella asiatica* showed S and G2/M arrest in human colon adenocarcinoma-derived Caco-2 cells, accompanied with the accumulation of cyclin B1 protein [50,51]. In addition, our results are in agreement with those of curcumin (diarylheptanoid derivative of turmeric), indicating that it inhibits cell proliferation by altering the cell cycle in different cancer cells [52,53].

## 4. Materials and Methods

### 4.1. Chemicals and Media

The growth medium RPMI 1640 containing 1 mmol/L L-glutamine, fetal bovine serum (FBS), phosphate buffered saline (PBS), and penicillin streptomycin were purchased from the American Type Culture Collection (ATCC) in Manassas, VA. The MTS assay kit was purchased from Promega Life Sciences (Madison, WI, USA). The lipid peroxidation, glutathione peroxidase, and catalase assay kits were purchased from Abcam (Cambridge, MA, USA). The comet assay kit was obtained from Trevigen (Gaithersburg, MD, USA). The propidium iodide was purchased from Calbiochem (La Jolla, CA, USA).

### 4.2. Vernonia calvoana Preparation and Fractionation

Leaves of *Vernonia calvoana* were harvested in Bangou, Cameroon. The leaves were rinsed and air-dried under the sun for a day. This was followed by shade-drying for 5 days. Five hundred grams of dried leaves were mixed with 600 mL of methanol and heated at 50 °C. The mixture was filtered with Whatman No 1 filter paper and evaporate to dryness using a rotary evaporator. *Vernonia calvoana* extracts were kept refrigerated at 4 °C until use. Fractionation of plant extract was performed according to Ogungbe et al. (2014) [54] and Abugri et al. (2016) [55]. The fractions were collected and stored in a freezer at −4 °C.

### 4.3. Cell Culture

Human ovarian adenocarcinoma (OVCAR-3) cells were purchased from the American Type Culture Collection. They were then sub-cultured in RPMI-1640 medium, supplemented with 10% fetal bovine serum and 1% penicillin/streptomycin (Thermo Scientific, Waltham, MA, USA), and grown in an incubator at 37 °C in 5% CO_2_. Fresh medium was supplemented every 48 h.

### 4.4. Cell Treatment and Determination of Cell Viability

The antiproliferative effects of VC F7 on the viability of OVCAR-3 cells was determined by the MTS colorimetric assay. We used fraction 7 because it showed the highest antitumor activity towards ovarian cancer cells compared to other fractions present in the crude extract. Cells were seated in 96-well plates at a density of 5 × 10^3^ cells per well were treated with different doses (0, 8, 16, and 32 μg/mL) of VC F7 for 48 h. After cells treatment, the medium was carefully aspirated from the treated plate and replace with an aliquot (100 µL) of fresh medium. Then, 20 µL of the MTS solution was added to each well and incubated for 3 h. The absorbance was read at 490 nm using a Biotex Model micro plate reader.

### 4.5. Morphological Changes

In this assay, we explore the morphological changes of OVCAR-3 cells treated and untreated with VC F7. Briefly, cells seated in each polystyrene 6 well-plate were treated with VC F7 for 48 h. After treatment, cells were washed twice with PBS and stained with a double dye (acridine orange (AO) and propidium iodide (PI). After staining, cells were examined and photographed under an Olympus fluorescent microscope.

### 4.6. Measurement of Lipid Peroxidation/Malondiadehyde

For this experiment, cells seeded in a 6-well plate at a density of 5 × 10^6^ cells/well were treated with different doses (0, 8, 16, and 32 µg/ mL) of VC F7 for 48 h. Cells were harvested, centrifuged, and collected in a 15 mL test tube. The cell pellets were lysed in 200 µL malondialdehyde (MDA) lysis buffer plus 2 µL BHT (100×) The freeze–thaw method was then carried out, and then 200 uL aliquots of the culture medium was assayed for MDA according to the lipid peroxidation assay protocol as previously described [56,57]. The absorbance was measured at 586 nm and the concentration of MDA was estimated from the standard curve. Experiments were performed in triplicates.

### 4.7. Measurement of Catalase Activity

Catalase activity was estimated by the means of catalase assay activity kit from Abcam company. Cells treated with different doses (0, 8, 16, and 32 µg/ mL) of VC F7 for 48 h. Treated and untreated cells were harvested, centrifuged, and collected in a 15-mL test tube. Cells were digested and catalase activity was determined according to the protocol as previously described with a few modifications [58]. Experiments were performed in triplicates.

### 4.8. Measurement of Glutathione Peroxidase Activity

To estimate the glutathione activity in this study, OVCAR-3 cells were seeded in a 6-well plate and treated with different doses of VC F7 for 48 h. Cells were digested and intracellular glutathione levels were determined using the assay kits purchased from Abcam (Cambridge, MA) according to the protocol previously described with some modifications [59]. Experiments were performed in triplicates.

### 4.9. Assessment of DNA Damage

For this experiment, OVCAR-3 cells were treated with VC F7 at doses of 8, 16, and 32 µg/mL for 48 h. Briefly, an aliquot 50 µL cell suspension was mixed with 200 µL of agarose. Then, 75 µL of the mixture was spread on the comet slides and place in the refrigerator for 15 min. The slides were immersed in lysis solution for 60 min at 4 °C. They were further immersed in a prepared alkaline solution and kept in the dark for 60 min. Sample slides were electrophoresed at 21 V for 30 min, dehydrated in 70% ethanol for 5 min, and stained with DNA- bound SYBR green I fluorescence stain overnight. The samples were visualized for DNA damage under a fluorescent microscope at 494/521 nm wavelength where several images were taken. The images were analyzed using the Trevigen Comet Assay software.

### 4.10. Assessment of Cell Cycle Distribution

All steps for this experiment were performed at 0 °C. Briefly, cells were seeded into a 6-well plates at the density of 6 × 10^6^ cells/well and treated with different doses (0, 8, 16, and 32 µg/ mL) of VC F7 for 48 h. After treatment, cells were harvested, washed twice with PBS, and fixed in ice-cold methanol for 30 min at 4 °C. Cells were stained with propidium iodide in the presence of RNase A and incubated for 30 min at room temperature. After staining, cells were analyzed by flow cytometry (BD Biosciences, San Jose, CA, USA).

## 5. Conclusions

New therapeutic approaches for the screening of bioactive compounds present in medicinal plants have received increasing attention due to their chemopreventive properties such as anti-oxidative, anti-cancer, and anti-inflammatory activities [60,61,62]. Studies indicated that *Vernonia calvoana* contained high concentrations of flavonoids rich in antioxidants which are natural substances that prevent cellular damage in living organisms. In the present study, we demonstrated that VC F7 is able to inhibit cell proliferation, induce DNA damage and cell cycle arrest at the S-phase checkpoint of the cell cycle in human ovarian cancer (OVAR-3) cells through oxidative stress, as demonstrated by an increase in MDA production and a decrease in catalase and glutathione activities in treated cells compared to the control. All these unique properties of VC F7 against OVAR-3 cells strongly suggest that VC F7 may be a novel and potential targeting molecule that can be used as a therapeutic agent for ovarian cancer. It is noteworthy that VC F7 exerts its anticancer activity by inhibiting cell proliferation, inducing DNA damage, and causing cell cycle arrest through oxidative stress in OVAR-3 cells. However, future preclinical and clinical trial studies are needed to test the medicinal properties of VC F7 as an alternative therapeutic agent for the prevention and/or treatment of ovarian cancer. Future research in our lab will focus on the identification and characterization of the active constituents present in CV F7 and testing antitumor activity in an animal model.

## Figures and Tables

**Figure 1 ijms-21-04429-f001:**
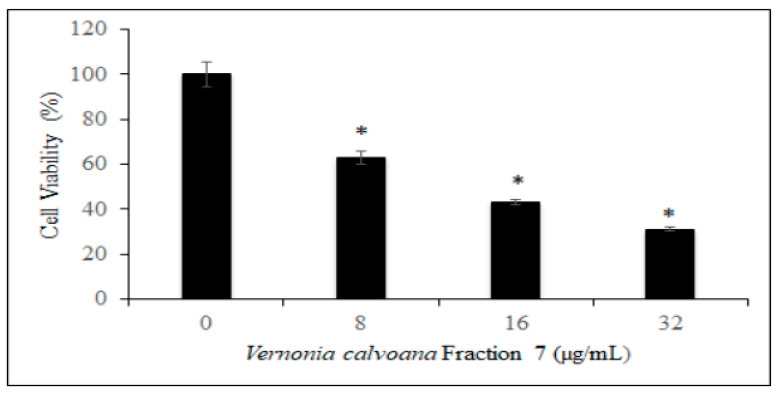
Antiproliferative effect of *Vernonia calvoana* fraction 7 (VC F7) on OVCAR-3 cells. The OVCAR-3 cells were treated with different doses of VC F7 for 48 h. The cell viability was measured by MTS assay as described in the Materials and Methods section. Each data point represents the mean value and standard deviation (*n* = 3). * Asterisks denote a statistically significant difference (*p* < 0.05) between the control and the treated groups according to the ANOVA Dunnett test.

**Figure 2 ijms-21-04429-f002:**
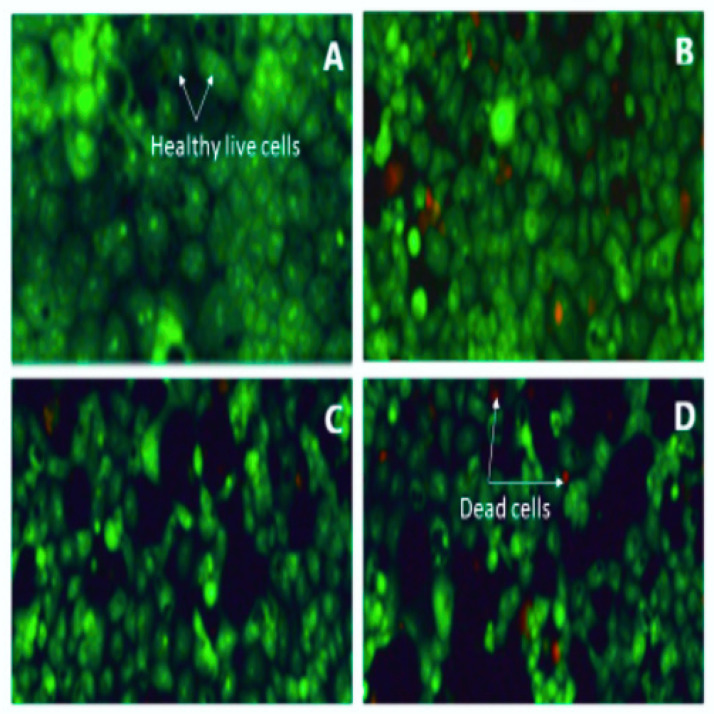
Representative fluorescence images of AO/PI-stained OVCAR-3 cells untreated and treated with VC F7. Live cells are stained in green and dead cells are stained red. (**A**)—Control, (**B**)—8 µg/mL, (**C**)—16 µg/mL, (**D**)—32 µg/mL. All fluorescence images were captured under 20× optical resolution of the microscope.

**Figure 3 ijms-21-04429-f003:**
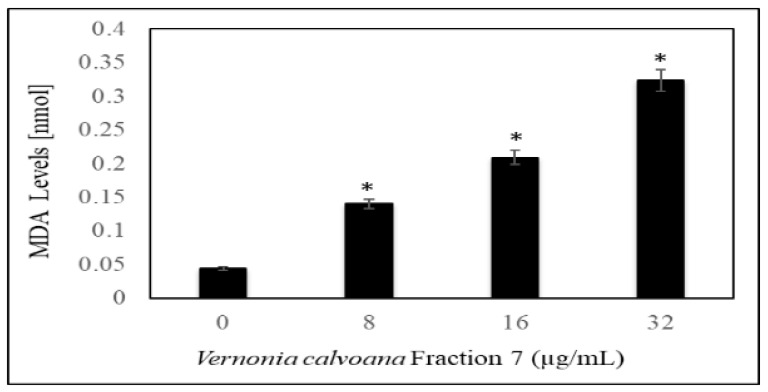
Effect of *Vernonia calvoana* fraction 7 (VC F7) on MDA production in untreated and treated OVCAR-3 cells for 48 h. The doses that were found to be statistically significantly different (*p* < 0.05) compared to the control are denoted by (*) according to ANOVA Dunnett.

**Figure 4 ijms-21-04429-f004:**
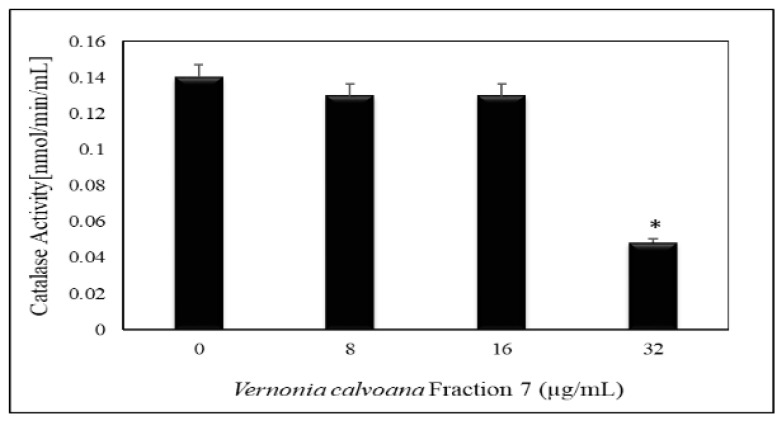
Effect of *Vernonia calvoana* fraction 7 (VC F7) on catalase activity in untreated and treated OVCAR-3 cells for 48 h. The doses that were found to be statistically significantly different (*p* < 0.05) compared to the control are denoted by (*) according to ANOVA Dunnett.

**Figure 5 ijms-21-04429-f005:**
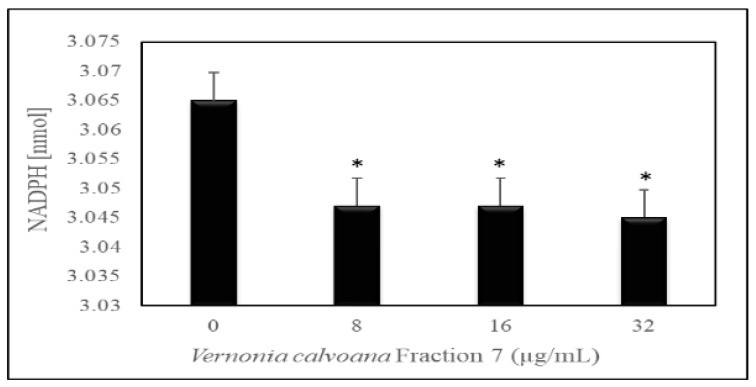
Effect of *Vernonia calvoana* fraction 7 (VC F7) on glutathione peroxidase activity in untreated and treated OVCAR-3 cells for 48 h. The doses that were found to be statistically significantly different (*p* < 0.05) compared to the control are denoted by (*) according to ANOVA Dunnett.

**Figure 6 ijms-21-04429-f006:**
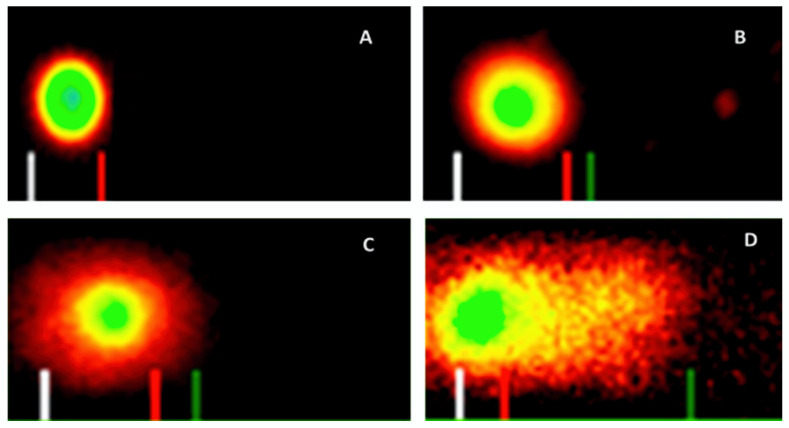
Representative SYBR green comet assay images of untreated (**A**-control) and *Vernonia calvoana* fraction 7 (VC F7)-treated cells at 8 µg/mL (**B**), 16 µg/mL (**C**), and 32 µg/mL (**D**) for 48 h.

**Figure 7 ijms-21-04429-f007:**
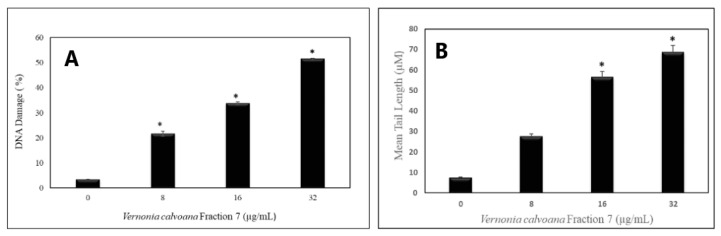
*Vernonia calvoana* fraction 7-induced DNA damage in OVCAR3 cells was measured by the comet assay. The OVCAR-3 cells were treated with different doses of VC F7 for 48 h. (**A**) represents the percentage of DNA cleavage and (**B**) represents the comet tail length. * *p* < 0.05 considered significant.

**Figure 8 ijms-21-04429-f008:**
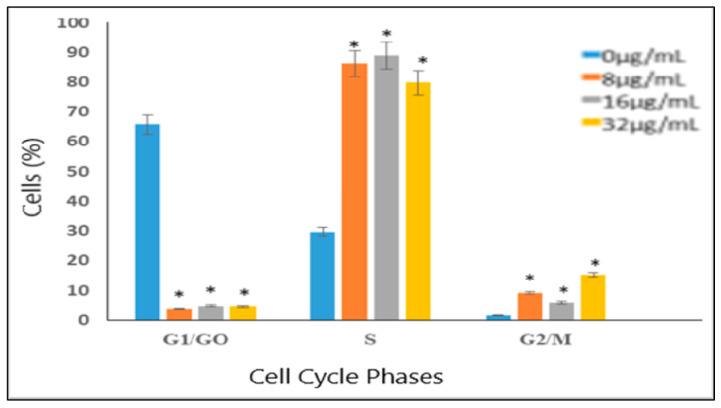
Bar graph showing percentage (%) of OVCAR-3 cells in different phases of the cell cycle. Each point represents the mean ± standard deviation of three independent experiments. * denotes statistically significant difference between the control and the treated group according to ANOVA (*p* < 0.05).

**Figure 9 ijms-21-04429-f009:**
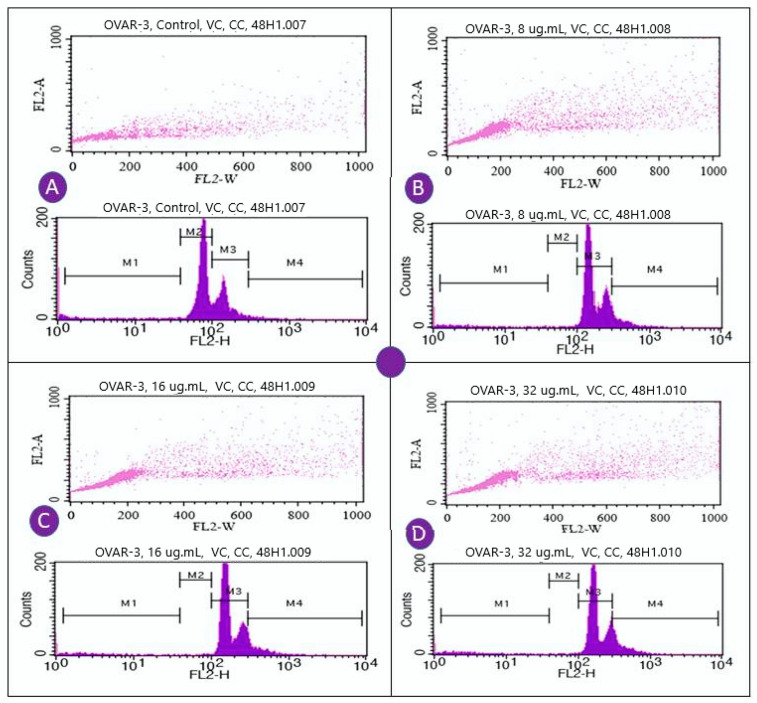
Representative dots plots and histogram showing cell cycle distribution of OVCAR-3 cells treated with VC-F7. The cells were fixed with methanol, stained with PI, and analyzed by flow cytometry (FACS Calibar; Becton-Dickinson) using CellQuest software as described in the Methods section. (**A**) 0 µg/mL, (**B**) 8 µg/mL, (**C**) 16 µg/mL, (**D**) 32 µg/mL. Three experiments were performed, and one (1) representative experiment is shown.

**Figure 10 ijms-21-04429-f010:**
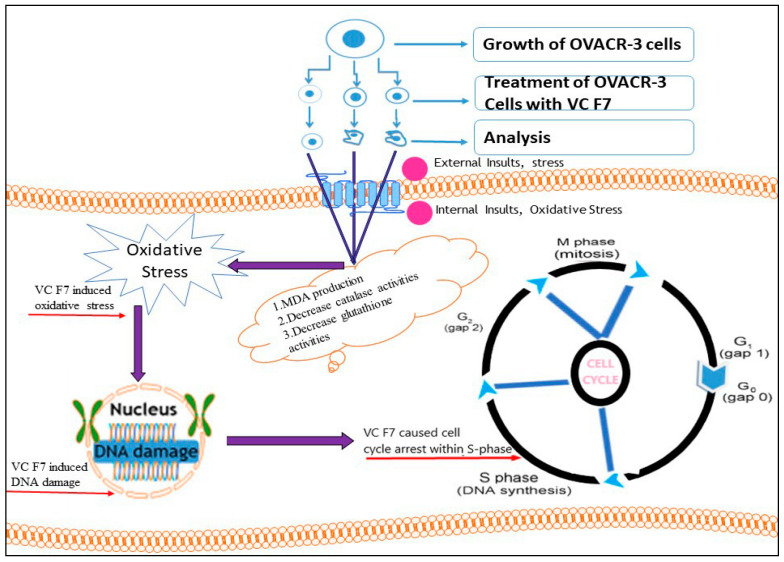
Underlying mechanisms by which *Vernonia calvoana* fraction 7 (VC F7) exerts its antitumor activity in human ovarian cancer (OVAR-3) cells.

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
