# Peer review of "Vernonia calvoana Shows Promise towards the Treatment of Ovarian Cancer"

_ijms, 2020, doi:10.3390/ijms21124429_

Round 1

Reviewer 1 Report

In the manuscript "Vernonia calvoana Shows Promise as Anticancer Agent towards the Treatment of Ovarian Cancer" why authour drying the leaves in sun? usually these kind of estracts will be dried in dark or shade dry in saving the extract or leaf from losing the active compounds in it. in 2.4 why the author specifically using the fraction 7? and the concentration 8,16 and 32 ug/ml?

When author mention  Vernonia calvoana fraction 7 (VC F7), no need to repeat it always and instaed it can be used as VC F7.

In Fig 2, it will be good if author can point out the difference with an arrow in it.

Manuscript with Vernonia calvoana in treating the ovarian cancer is a appreciable study as the modern medicine gives a side effect and these kind of medicinal plant treat the disease with very low side effect  an sometimes not.  When saying it as dietary intake how these medicinal plants can be useful?  Whether the author has any idea to test this compound in the animal model as the invitro alone cannot be enough to prove the effect of a medicinal plant in ovarian cancer

Author Response

Dear Reviewer,

We would like to thank you for the thoughtful feedback and helpful comments.  To address the issues that were raised, we have streamlined and focused the manuscript considerably.  We have also gathered and compiled additional data/information so the changes that have been made are substantial and are intended to address all of the issues raised. We trust that you will find that this is a significant improvement to the review. In the marked-up manuscript, changes have been highlighted in blue and modified texts are underscored.

Reviewer: In the manuscript "Vernonia calvoana Shows Promise as Anticancer Agent towards the Treatment of Ovarian Cancer" why authors drying the leaves in sun? usually these kind of extracts will be dried in dark or shade dry in saving the extract or leaf from losing the active compounds in it. in 2.4 why the author specifically using the fraction 7? and the concentration 8, 16 and 32 μg/mL?

Response to Reviewer Comment: We thank the reviewer for his/her helpful comments. (1) We added more information about preparation of extract related to the drying process: “The leaves were rinsed and air-dried under the sun for a day. This was followed by shady dry for 5 days.” (2) We used fraction 7 because it showed the highest antitumor activity towards ovarian cancer cells compared to other fractions present in the crude extract. To refer to this point, we added the following text: “We used fraction 7 because it showed the highest antitumor activity towards ovarian cancer cells compared to other fractions present in the crude extract”. (3) We selected the doses of 8, 16 and 32 μg/mL based on their proximity to inhibition dose (IC50 = 18.56 μg/mL). Our intent was to explore different concentrations with potential for dosing in future studies.

Reviewer: When author mentions Vernonia calvoana fraction 7 (VC F7), no need to repeat it always and instead it can be used as VC F7.

Response to Reviewer Comment: We agree with this critique and used VC F7 through the text as part of our overall effort to streamline the manuscript.

Reviewer: In Fig 2, it will be good if author can point out the difference with an arrow in it.

Response to Reviewer Comment:  We thank the reviewer for the thoughtful feedback. We added the arrow on Fig 2 to show the difference between live and dead cells.

Reviewer: Manuscript with Vernonia calvoana in treating the ovarian cancer is an appreciable study as the modern medicine gives a side effect and these kind of medicinal plant treat the disease with very low side effect and sometimes not.  When saying it as dietary intake how these medicinal plants can be useful?  Whether the author has any idea to test this compound in the animal model as the in vitro alone cannot be enough to prove the effect of a medicinal plant in ovarian cancer

Response to Reviewer Comment:  We thank the reviewer for the thoughtful feedback. Future research in our lab will be focusing of the identification and characterization of the possible constituents (secondary metabolites) present in CV F7 and testing its antitumor activity in an animal model. To refer to this comment, we added to following text to the revised manuscript: “Future research in our lab will focus on the identification and characterization of the active constituents present in CV F7 and testing antitumor activity in an animal model.”

Reviewer 2 Report

The manuscript entitled “Vernonia calvoana Shows Promise towards the Treatment of Ovarian Cancer” reports an interesting study of cytotoxic activity against cancer cells of 10 fractions from the crude extract of Vernonia calvoana. Additionally, the authors studied the hypothesis of the mechanism of the extract activity. The study can be used in further studies, since the results show the fraction of Vernonia calvoana can inhibit cancer (OVAR-3) cells proliferation, inducing DNA and causing cell cycle arrest through oxidative 3 stress in OVAR-3 cells.

There are some points that can improve the manuscript.

1 - The abstract is a little bit longer.

Please shortened the abstract.

2 – Abstract – page 1 – line

“antitumor activity towards ovarian cancer cells”

Please, I think that will increase the visibility of the manuscript if the authors add the type of cancer cell:

“antitumor activity towards ovarian cancer cells (OVCAR-3)”

3 – Figures 1, 3, 4, 5, 7–

Please improve the figures. It is hard to visualize the standard deviation. Perhaps changing the color of the standard deviation.

4 – Figure 8

Please improve the quality of the labels of figure 8

5 – I think that would improve a lot the visibility of the manuscript if the authors added some insights into the possible constituents (secondary metabolites) present in fraction 7. Or even some preliminary differences on the fraction7 from the other fractions using some simple spectral data as NMR or mass spectra. Even more, using some simple multivariate approach as PCA.

Author Response

Dear Reviewer,

We would like to thank you for the thoughtful feedback and helpful comments.  To address the issues that were raised, we have streamlined and focused the manuscript considerably.  We have also gathered and compiled additional data/information so the changes that have been made are substantial and are intended to address all of the issues raised. We trust that you will find that this is a significant improvement to the review. 

Reviewer: The manuscript entitled “Vernonia calvoana Shows Promise towards the Treatment of Ovarian Cancer” reports an interesting study of cytotoxic activity against cancer cells of 10 fractions from the crude extract of Vernonia calvoana. Additionally, the authors studied the hypothesis of the mechanism of the extract activity. The study can be used in further studies, since the results show the fraction of Vernonia calvoana can inhibit cancer (OVAR-3) cells proliferation, inducing DNA and causing cell cycle arrest through oxidative 3 stress in OVAR-3 cells.

There are some points that can improve the manuscript.

  1. The abstract is a little bit longer.

Please shortened the abstract.

  1. Abstract – page 1 – line

“antitumor activity towards ovarian cancer cells”

Please, I think that will increase the visibility of the manuscript if the authors add the type of cancer cell:

“antitumor activity towards ovarian cancer cells (OVCAR-3)”

  1. Figures 1, 3, 4, 5, 7–

Please improve the figures. It is hard to visualize the standard deviation. Perhaps changing the color of the standard deviation.

  1. Figure 8

Please improve the quality of the labels of figure 8

  1. I think that would improve a lot the visibility of the manuscript if the authors added some insights into the possible constituents (secondary metabolites) present in fraction 7. Or even some preliminary differences on the fraction 7 from the other fractions using some simple spectral data as NMR or mass spectra. Even more, using some simple multivariate approach as PCA.

Response to Reviewer Comment: We thank the reviewer for the thoughtful feedback and helpful comments. To address the concern that has been raised, we have done the following:

  1. We shortened the length of the abstract.
  2. We added “OVCAR-3” to ovarian cancer cells when using them in the text to increase visibility as suggested by the reviewer.
  3. We improved the quality of figures 1, 2, 3, 4, 5, 7 and 8.
  4. Future research in our lab will be focusing of the identification and characterization of the active constituents present in fraction 7 and testing its antitumor activity in animal model. To refer to this comment, we added to following text to the revised manuscript: “Future research in our lab will focus on the identification and characterization of the active constituents present in CV F7 and testing antitumor activity in an animal model.”.